# Low carbon renewable natural gas production from coalbeds and implications for carbon capture and storage

Zaixing Huang[1,2], Christine Sednek[1], Michael A. Urynowicz[1,2], Hongguang Guo[3], Qiurong Wang[4], Paul Fallgren[5], Song Jin[2,5], Yan Jin[1], Uche Igwe[1] & Shengpin Li[1]

Isotopic studies have shown that many of the world's coalbed natural gas plays are secondary biogenic in origin, suggesting a potential for gas regeneration through enhanced microbial activities. The generation of biogas through biostimulation and bioaugmentation is limited to the bioavailability of coal-derived compounds and is considered carbon positive. Here we show that plant-derived carbohydrates can be used as alternative substrates for gas generation by the indigenous coal seam microorganisms. The results suggest that coalbeds can act as natural geobioreactors to produce low carbon renewable natural gas, which can be considered carbon neutral, or perhaps even carbon negative depending on the amount of carbon sequestered within the coal. In addition, coal bioavailability is no longer a limiting factor. This approach has the potential of bridging the gap between fossil fuels and renewable energy by utilizing existing coalbed natural gas infrastructure to produce low carbon renewable natural gas and reducing global warming.

[1] Department of Civil and Architectural Engineering, University of Wyoming, Laramie, Wyoming 82071, USA. [2] Center for Biogenic Natural Gas Research, University of Wyoming, Laramie, Wyoming 82071, USA. [3] College of Mining Technology, Taiyuan University of Technology, Taiyuan 030024, China. [4] Department of Animal Science, University of Wyoming, Laramie, Wyoming 82071, USA. [5] Advanced Environmental Technologies LLC, Fort Collins, Colorado 80525, USA. Correspondence and requests for materials should be addressed to Z.H. (email: zhuang@uwyo.edu) or to Q.W. (email: qwang@uwyo.edu)

The Powder River Basin (PRB), located in Wyoming and Montana, is the largest low-sulfur subbituminous coal deposit in the world, supplying approximately 40% of the coal utilized in the United States[1, 2]. The PRB contains vast layers of more than 50 prominent coal seams amounting to 1.07 trillion short tons of in-place resources, yet only 25 billion are estimated to be economically recoverable. This leaves approximately 98% of the known coal resource unminable; however, these resources have been exploited for natural gas[1, 2]. The PRB has coalbed natural gas (CBNG) reserves ranging between 7 and 39 trillion cubic feet (between 0.20 and 1.1 trillion m$^3$)[3–5]. The PRB has been one of the world's most active areas for CBNG production, where CBNG wells in the PRB increased from fewer than 1000 in 1999 to ~25,000 (with 8000 shut-in wells) in 2010[5–7]. In the late 1990s, the estimated well life was 12–17 years for the Wyodak coal zone[6]. More recent empirical data from wells in the Wyodak and Big George coal zones has shown an average of 8 years of production with the average peak production rate typically observed after 1.2 years[5]. Rates then declined at a rate of approximately 45% per year. Consequently, enhancing CBNG production from depleted wells is a topic of great interest for both commercial and governmental sectors.

Isotopic analyses have indicated that the methane (main constituent in CBNG) in the PRB coal seams is primarily biogenic in origin[3, 5]. Physical and geological characteristics of the PRB including, surface area and pores for attached microbial growth, mesophilic temperatures, moisture, and near neutral pH[8, 9], have been shown to support microbial activities. The Wyodak coal zone also has high permeability ranging up to 1 Darcy[3], which adds in resource extraction and nutrient flushing for biogenic processes. Temperature conditions, from 16 to 38 °C (61 to 100°F) and pressures, from 689 to 758 kPa (100 to 110 psi)[10] are well within the ranges where many methanogenic communities thrive. These properties and characteristics suggest that CBNG production in the PRB may be enhanced by stimulating the microbial activities within the coal seam. In fact, several approaches have been attempted commercially[8, 11] including biostimulation with nutrients and bioaugmentation[11, 12]; however, these methods are limited to bioavailable carbon content of the coal[13, 14].

Based on the mechanisms of the gas formation, a concept of low carbon renewable natural gas (LCRNG) may be formulated as depicted in Fig. 1. Coal seams can be envisioned as geobioreactors that can be engineered for the enhanced production of biogenic gas. The process is similar to in situ biodegradation where an alternative carbon source(s) and/or nutrients are added to enhance the degradation and transformation of contaminants of concern (COCs). However, in this case the objective is not groundwater remediation but the production of large quantities of LCRNG using an alternative substrate such as simple carbohydrate compounds derived from locally grown perennial forage crops, such as alfalfa, switch grass, and miscanthus, or sugar beet. Unlike anaerobic digester (AD) gas, it benefits from the use of existing CBNG infrastructure (wells, pipelines, compressors, etc.), substantially reducing the cost of producing the gas while achieve a massive scale that is infeasible for conventional AD. The natural gas produced by this approach can be considered carbon neutral or even carbon negative, as carbon not directly converted to methane (i.e., microbial biomass and carbon dioxide) may be sequestered within the coal[15–19]. That is carbon dioxide in the atmosphere is fixed by photosynthesis to produce biomass. The biomass such as sugar beet is processed to produce simple carbohydrates that are soluble at aboveground facilities and injected into the coal seams as a solution. These carbohydrates are subsequently transformed to natural gas and stored in the coal seams. Coal has a very large surface area (greater than 800 m$^2$ g$^{-1}$ coal for Canyon coal[20]) and a strong affinity to adsorb gases. Since much of the injected carbon remains in the form of microbial biomass and carbon dioxide, the carbon intensity of the LCRNG is further lowered as a result of carbon capture and storage. In this study, monosaccharaides (five-carbon and six-carbon sugars) and a disaccharide (cellobiose) are chosen to evaluate the gas production potential using native coal and the consortium of microorganisms living within the coal as the inoculum source for the bench-scale experiments. The impact on the energy content of coal is also investigated. The results suggest that microorganisms indigenous to coal can convert plant-derived carbohydrates to natural gas and coalbeds can act as natural geobioreactors to produce low carbon renewable natural gas without lowering the energy content of coal. To our knowledge, this is the first study to address the potential for the production of LCRNG from coal seams by using an alternative substrate.

## Results

**Yield and rate of methane production.** As shown in Fig. 2, the monosaccharide (five- and six-carbon carbohydrates) and acetic acid-fed microcosms produced significant amounts of methane. Particularly, over 7000 μmol g$^{-1}$ carbon substrate of methane was formed for five-carbon sugars. Cellulobiose produced a minimal amount of methane. The positive control with hydrogen and bicarbonate as an electron acceptor produced minimal amount of methane. The negative control had no detectable methane throughout the entire incubation period.

The microbes in the coal microcosms containing monosaccharides (glucose, galactose, mannose, arabinose, and xylose) could metabolize the sugars to more substantially than those containing disaccharide (cellobiose), forming significant amount of methane. Since carbohydrates were not detected following Day 2, the indigenous coal culture was capable of rapidly metabolizing the alterative carbon sources (glucose, galactose, mannose, xylose, and arabinose), suggesting a high affinity to the plant-derived monosaccharides. Cellobiose was not detected following Day 46,

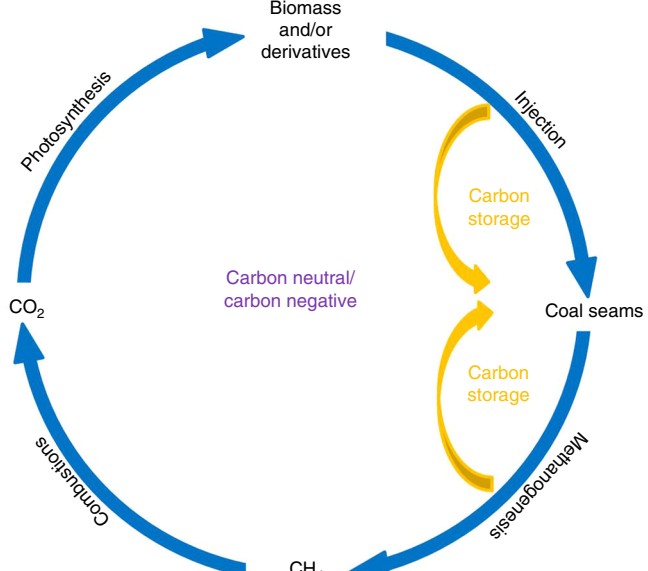

**Fig. 1** Schematic depicting the concept of low carbon renewable natural gas produced using coalbeds as geobioreactors and as sinks for carbon capture and storage. The concenpt is intended to take advantage of the coal seams as natural geobioreactors that use biomass derived simple sugars as the carbon source for natural gas production. Coal seams also serve as a sink for sequestering microbial biomass and carbon dioxide, further lowering the carbon intensity of the gas

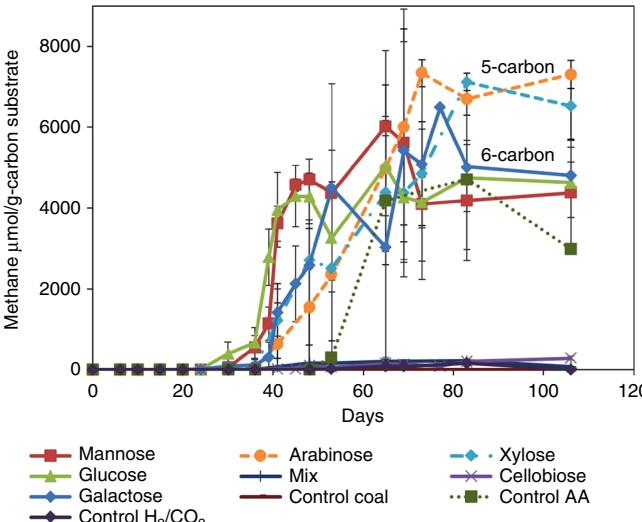

**Fig. 2** Profile of methane yeilds in all microcosms. The highest yeilds were from galactose, mannose, glucose, xylose, arabinose, and acetic acid-fed systems. The lowest yeilds were from the mix, cellobiose, $H_2/CO_2$, and controls with only coal. Data points represent the means ± s.d., $n = 3$ per group

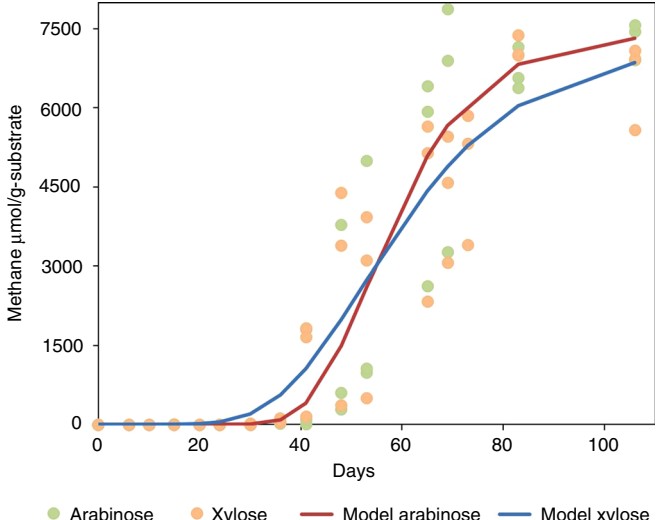

**Fig. 3** Gompertz model and raw data points for the five-carbon carbohydrates. The five-carbon carbohydrates include arabinose and xylose. The values of the asymptote for methane Arabinose A = 7339 and xylose A = 7124, the parameters $\mu_m$, and $\lambda$ are displayed in Table 1

**Table 1 Methane production kinetic parameters for a 128-day incubation period of anaerobic microcosms inoculated with PRB coal**

| Treatment/Control | A ($\mu$mol g$^{-1}$ carbon) | $\mu_m$ ($\mu$mol g$^{-1}$ carbon per day) | $\lambda$ (day) | $R^2$ |
|---|---|---|---|---|
| *Monosaccharide six-carbon carbohydrates* | | | | |
| Galactose | 6488 (±326) | 303 | 38 | 0.75 |
| Mannose | 6019 (±325) | 1375 | 38 | 0.83 |
| Glucose | 5041 (±828) | 805 | 35 | 0.92 |
| *Monosaccharide five-carbon carbohydrates* | | | | |
| Xylose | 7124 (±220) | 151 | 35 | 0.86 |
| Arabinose | 7339 (±332) | 232 | 42 | 0.87 |
| *Disaccharide cellobiose* | | | | |
| Cellobiose | 218 (±48) | 1.9 | 53 | 0.56 |
| *Mix and controls* | | | | |
| Mix | 286 (±57) | 1.8 | 60 | 0.24 |
| Negative control | 0 | 0 | — | — |
| Acetic acid control | 4713 | 246 | — | — |
| $H_2/CO_2$ control | 158 | 0.006 | 65 | 0.0445 |

suggesting that the degradation of the more complex disaccharide cellobiose required additional time.

Lag times were generally between 30 and 40 days for the carbon-fed treatments (See Fig. 2). The acetic acid-fed system had a longer lag period of approximately 65 days which was unexpected, considering that acetate is the primary substrate for acetoclastic methanogenesis. Methanogens that use acetate such as *Methanosarcina* were shown to have low populations and diversity ("Microbial community analysis" Section). It is possible that community selection and dynamics established in the monosaccharide containing microcosms represented a more natural transition to methanogenic conditions than those represented by the acetate control, thereby resulting in shorter lag times.

Table 1 summarizes the rates, yields, and lag times for the treatments and controls. Sigmoidal Gompertz model was used to estimate five- and six-carbon carbohydrates while cellobiose, mix

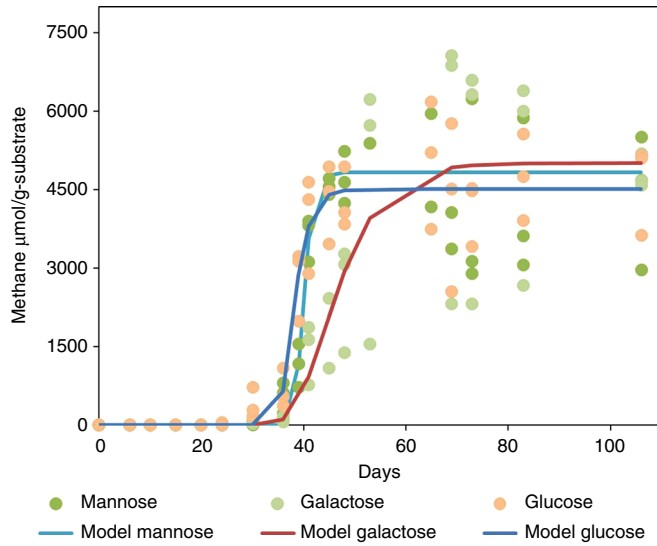

**Fig. 4** Gompertz model and raw data points of the six-carbon carbohydrates. The six-carbon carbohydrates include mannose, galactose, glucose. The values of the peak methane galactose A = 6488, mannose A = 6019 and glucose A = 5041, the parameters $\mu_m$, and $\lambda$ are displayed in Table 1

and the $H_2/CO_2$ controls were modeled using linear model. The highest rate of methane was from the six-carbon carbohydrates, with mannose being the highest, at 1375 $\mu$mol g$^{-1}$ carbon day$^{-1}$. This constitutes towards a rapid methane production where within 5 days the peak methane was formed.

Figures 3 and 4 show the five-carbon (arabinose and xylose) and six-carbon (glucose, galactose, and mannose) carbohydrate treatments. Based on the Gompertz modeling, it appears that both five-carbon and six-carbon carbohydrates performed in a similar manner. To further view the differences between the three structural types of carbohydrates used in this investigation, the five- and six-carbon carbohydrates and cellobiose were consolidated individually. Fig. 5 shows the cumulative methane production rates for the consolidated five-carbon carbohydrates, six-carbon carbohydrate treatments and cellobiose and the rate

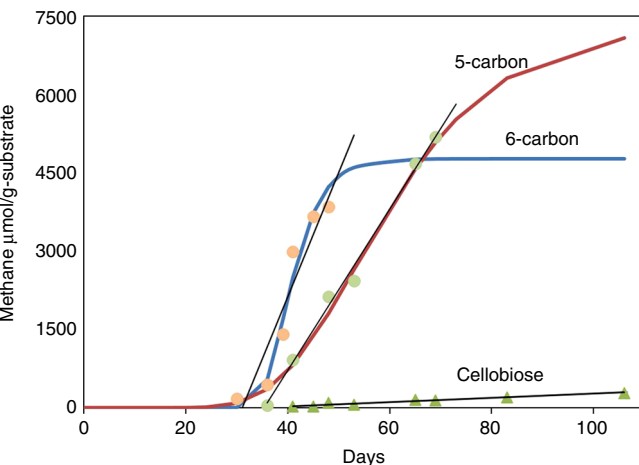

**Fig. 5** Linear methane rates and Gompertz model production curves of monosaccharides and disaccharides. The *blue curve* is the averaged six-carbon carbohydrates and the *red curve* is the averaged five-carbon carbohydrate treatments. The *black line* with *green triangle* data points represents the cellobiose treatments

differences between the individually consolidated five-carbon carbohydrates and six-carbon carbohydrates and cellobiose treated coal. The five-carbon carbohydrate treatment rate was slower with a linear rate of ~155 $\mu$mol g$^{-1}$ carbon with strong correlation ($R^2 = 0.993$). The six-carbon carbohydrate treatment rate was highest with ~238 $\mu$mol g$^{-1}$ carbon with correlation ($R^2 = 0.887$). The cellobiose carbohydrate-fed treatments showed less methane production, which were modeled linearly, having rates of ~4 $\mu$mol g$^{-1}$ cellobiose and strong correlation ($R^2 = 0.949$).

**Conversion efficiency**. To understand the performance of the bench microcosms, a stoichiometric equation was used for anaerobic methanogenic treatment[21]. This equation accounts for the mass balance and, more importantly, the electron balance of the conversion of the carbohydrates to methane. Maintenance and growth energy factors were assumed to be fs = 0.2 and fe = 0.8, and that the biodegradability of the compound was 100% (which confirmed by the intermediates study). The calculations were converted into STP (1 atm, 25 °C). Conversion efficiencies were calculated according to the method of Owen et al. (1979)[22], where the peak methane was compared to the theoretical amount of methane.

In Supplementary Table 1, the conversion efficiencies for the peak methane production of each triplicate were averaged and assessed by using the Rittman and McCarty equation and the Owen method[21, 22]. The conversion efficiencies for the five- and six-carbon carbohydrates were statistically the same for the peak methane productions. A one-way analysis of variance (ANOVA) was used to test for conversion efficiency differences amongst the carbon sources using an $\alpha = 0.05$. Conversion efficiencies differed significantly across the carbon sources, with an F-value (2, 18) = 9.38, and a $p$-value > 0.0001. Least significant difference (LSD) post hoc comparisons of the carbon sources indicate that the five- and six-carbon carbohydrates were statistically the same where mean values ranged 42–55%, LSD = 23.1%, and the carbon-fed treatments formed significantly higher conversion efficiencies than the coal ($M = 0$, 95% range [0,0]).

**Intermediate organic acids and pH**. The pH values and some of the intermediate organic acids results are shown in Fig. 6. Lactic acid, formic acid and acetic acid were detected in all treatments during the incubation period with quantifiable amount while

acids such as propionic, ethyl acetate, and iodoacetic acid also showed presence in trace amounts. Formic acid and acetic acids are good precursors and can be utilized by hydrogentrophic and acetoclastic methanogens, respectively to produce methane[23]. Lactic acid can be readily transformed to methanogenic substrates including formic acid, acetic acid, methanol and $CO_2$[24]. The solid lines with filled circles represent the summation of the three organic acids. The trend of the summed organic acids started with a relatively high concentration at Day 2 (corresponding to the catabolism of the added substrate) that decreased during the incubation when methane started to form and accumulated towards the end of the incubation period, during which the gas production reached the plateau. The yield of acetic acid from the carbohydrates was in the range between carbohydrate structure groups five-carbon carbohydrates (1.125–1.148 mol mol$^{-1}$ substrate) and six-carbon carbohydrates (0.971–1.503 mol mol$^{-1}$ substrate); and cellobiose had the highest acetic acid yield of 1.928 mol mol$^{-1}$ cellobiose. Strong negative correlation between acetic acid concentration of the microcosms and the production of methane were indicated by the Pearson correlation matrix statistical testing.

The optimal pH values for methanogenic growth are between 6.5 and 7.2, with greater concern for pH lower than 5.5[25]. The pH is one of the major parameters controlling methanogenesis in anaerobic digester applications along with temperature[21, 26, 27]. The intention of the experimental design was to have enough buffer capacity in order to maintain the pH around 7, but the 1 mg l$^{-1}$ (NaHCO$_3$) buffer was not sufficient and the pH of all the carbohydrate-fed microcosms were below 6.5. The pH for the controls remained within the optimal range for methanogenesis. The five- and six-carbon carbohydrate-fed treatments were slightly lower, starting at pH between 5.7 and 6.0 on Day 2, presumably due to the rapid degradation of carbohydrates and the generation of intermediate organic acids (also indicated in Fig. 6). The pH of the five- and six- carbon carbohydrate treatments increased to the range of 6.3 to 6.5 over time for the rest of the incubation period. From Day 46 until the end of the incubation period pH was within the favorable range for methanogenesis (6–8)[25], but not always optimal. In the cellobiose and mix treatments that contained cellobiose, the pH values remained low throughout the incubation period, attributing to the slightly acidic conditions that occur when cellobiose dissolves in water, and the formation of organic acid during incubation.

**Carbon balance**. To assess the anaerobic microbial metabolisms in this study, a carbon balance was constructed. The carbon mass balance on the microcosms assumes: (1) the added substrate was completely consumed; (2) acetic, formic, and lactic acids were the dominant intermediate compounds; (3) the dominant gases were carbon dioxide and methane; (4) the aqueous chemistry of the carbonate system occurring within the microcosms was accounted for by the mass and charge balances of the bicarbonate buffer added, pH, and speciation; (5) the equations to estimate the carbon speciation assumes a steady state; (6) the rest of non-accounted carbon was deposited as a form of biomass; and (7) the adsorption of any form of carbon (except biomass) onto coal surface was negligible. The carbon added in the microcosms reside in biomass, organic acids and alcohols, $CH_4$, biogenic $CO_2$, $HCO_3^-$, $H_2CO_3$, or $CO_3^{2-}$, the original carbohydrate, and/or the coal.

Cell biomass estimation was done using the Rittmann and McCarty (2001)[21] Eq. (1).

$$Y = \frac{f_s^o \left[ M_c \frac{\text{g cells}}{\text{mol cells}} \right]}{\left[ \frac{n_e \, \text{e}^- \text{eq}}{\text{mol cells}} * \frac{8\text{g COD}}{\text{e}^- \text{eq donor}} \right]} \quad (1)$$

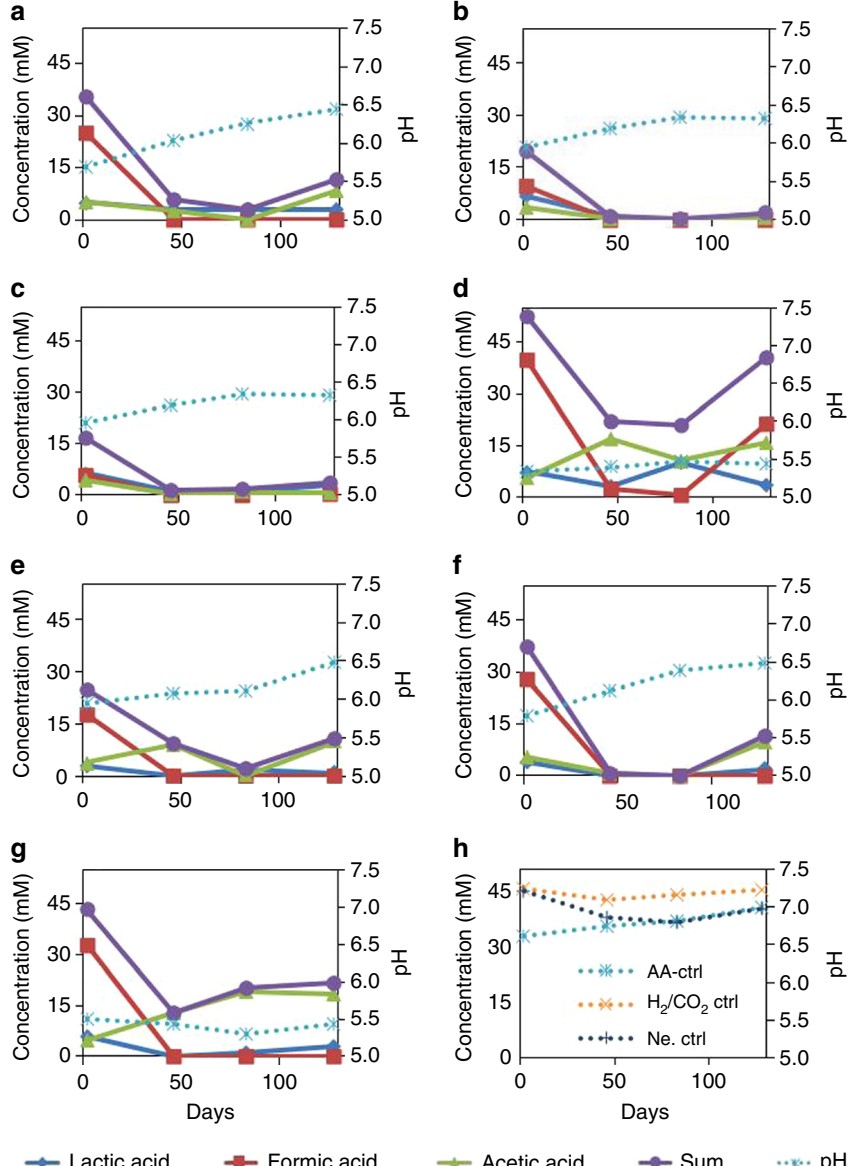

**Fig. 6** Intermediate organic acids and pHs at sampling intervals. The sampling intervals are day 2, 46, 83 and 128. Each data point represents an average value of three replicates. **a** Galactose; **b** mannose; **c** glucose; **d** cellobiose; **e** xylose; **f** arabinose; **g** mix; **h** controls (pH only)

where $M_c = 113$ g cells per mol cells (general formula for biomass $C_5H_7O_2N$), $n_e = 20$ e$^-$eq mol$^{-1}$ cells, $Y = 0.706f_s$, and $f_s = 0.2$[21]; assuming steady state conditions, neglecting decay and ammonium ($NH_4^+$) as the nitrogen source.

The carbon balance of six- and five- carbon treatments over time were demonstrated in Figs. 7 and 8. Day zero column shows the added carbohydrate amount stacked on top of the added carbon from the buffer. Days, 2, 46, 83, and 106 describe the summation of the various forms the carbon transformed into. The detected forms of carbon were formic, acetic, and lactic acids in the liquid portion of the culture, and in the gaseous phase, methane and carbon dioxide. The carbonate speciation of the system is sensitive to pH changes. Carbon dioxide from pH and speciation was subtracted from the total carbon dioxide detected. Hypothetically, carbon, hydrogen, and oxygen in the carbohydrates can be transformed to methane, carbon dioxide, and biomass, despite many pathways involved. However, this simplified calculation may overestimate the values of biomass because there might be carbon unaccounted in the batch systems. The true yields must be lower.

**Energy content analysis**. Supplementary Table 2 shows the average ash, sulfur, and heating values of the treatments. The average ash ranged from 9.27 to 11.78% by weight, higher than typical ash of 6.18–3.50% of the PRB coal[4]. The sulfur content had low variation between treatments and was approximately 0.3% by weight, within the range of typical PRB coal (0.25–0.525)[4]. The heating values varied amongst replicates, ranging from 11,393 to 11,922 Btu lb$^{-1}$, comparable with the subbituminous classification of the PRB coal 8000 to 12,000 Btu lb$^{-1}$ [4]. The six-carbon carbohydrates treated coal was found to have the highest Btu lb$^{-1}$ values on average, followed by the incubated control and the five-carbon carbohydrates treated coal. The non-incubated control has the lowest heating value. The data shows approximately 3% difference between the lowest treatments (non-incubated control) and the maximum treatment (6-carbon).

A one-way ANOVA was used to test for heating value differences among the four treatments: six-carbon, five-carbon, incubated control, and non-incubated control with an $\alpha = 0.05$. With equal variances, max/min ratio [24.75], statistical threshold [39.2][28]. Heating values differed significantly across the four

treatments F (3, 12) = 64.99, $p > 0.0001$. Five- and six-carbon carbohydrates treated samples were statistically different from the incubated control with heating values 11,463, 11,857, 11,682 Btu lb$^{-1}$, respectively. The heating values of the incubated

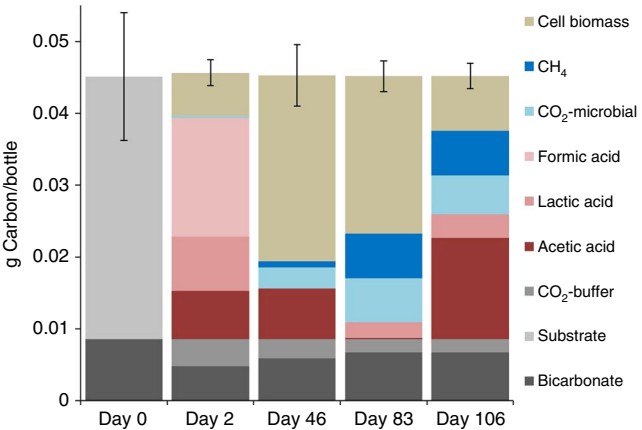

**Fig. 7** Carbon balance of five-carbon microcosms at sampling intervals. The sampling intervals are day 2, 46, 83 and 128. The columns represent the cumulative carbon levels of various compounds. Day 0 represents the total carbon (g per bottle) initially added to the system. Data points represent the means ± s.d., $n = 3$ per group

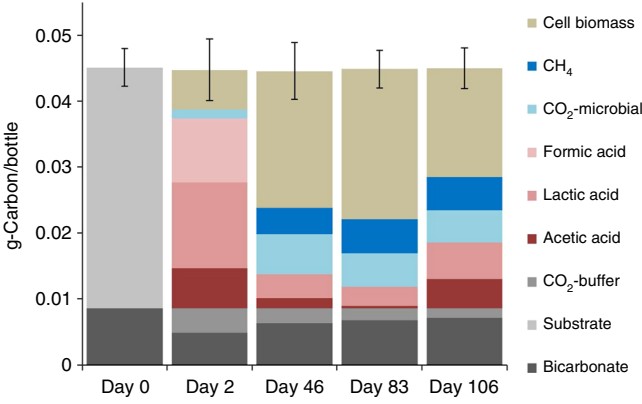

**Fig. 8** Carbon balance of six-carbon microcosms at sampling intervals. The sampling intervals are day 2, 46, 83 and 128. The columns represent the cumulative carbon levels of various compounds. Day 0 represents the total carbon (g per bottle) initially added to the system. Data points represent the means ± s.d., $n = 3$ per group

samples are all statistically higher than the non-incubated controls. These results suggest that the carbohydrate-fed incubations did not lower the Btu lb$^{-1}$ of coal, rather it appears to have increased the coal energy content slightly, presumably attributing to some biomass growth and biotransformation of the coal biomass mixture.

Bio-stimulating coalbed natural gas wells with nutrients has raised concerns about the loss of coal energy; however, incubation and biostimulation of the coal increased the coal energy content slightly within this investigation. The five-carbon carbohydrate was statistically the same as the non-incubated control. Still, this preliminary investigation of the energy content of the coal showed that the carbohydrate-fed incubations did not lower the heating values, when compared to a non-carbohydrate supplied coal.

**Microbial community analysis.** A total of 23,054 sequence reads with 678 OTUs at 97% similarity were obtained after sequencing and quality control. It consisted of only three (0.01%) archaeal sequence reads with two OTUs. The two archaeal OTUs were classified as methanogens, which included the genera *Methanosarcina* and *Methanoregula*. *Methanosarcina* metabolizes a wide variety of substrates including H$_2$, acetate, methanol, and methylamine[29]. *Methanoregula* are primarily obligate hydrogenotrophs, using H$_2$/CO$_2$ for growth and methane production[30]. The limited archaeal diversity may have been due to the lack of niches for microorganisms in coal to inhabit. Other studies have demonstrated the difficulties of amplifying archaeal 16S rRNA gene sequences from coal samples[31]; however the primer set 515F/806R is known to cover methanogens in wastewater treatment[32] and diverse archaeal communities in coal seams in China[33]. The low archaeal sequence reads may indicate low concentrations of methanogens from the samples, which may result in the observed long lag phase of gas production (Fig. 2).

Proteobacteria was the dominant in bacterial community, accounting for 67.01% of sequence reads, which was consistent with previously reported microbial diversity within CBNG fields[34]. Sequence reads of 20.61 and 10.81% were classified as Actinobacteria and Bacteroidetes, respectively, while other phyla contained <1.00% of sequence reads. The composition of bacterial communities is summarized in Fig. 9. And a phylogenetic tree was constructed with reference sequences as shown in Supplementary Fig. 1. High diversity of bacterial genera was detected from the coal sample where six genera had sequence read greater than 5%. These included *Bradyrhizobium* (19.53%), *Rhodopseudomonas* (8.24%), and *Pseudolabrys* (6.09%) in Proteobacteria, *Arthrobacter* (10.81%) and *Nocardioides* (8.06%) in

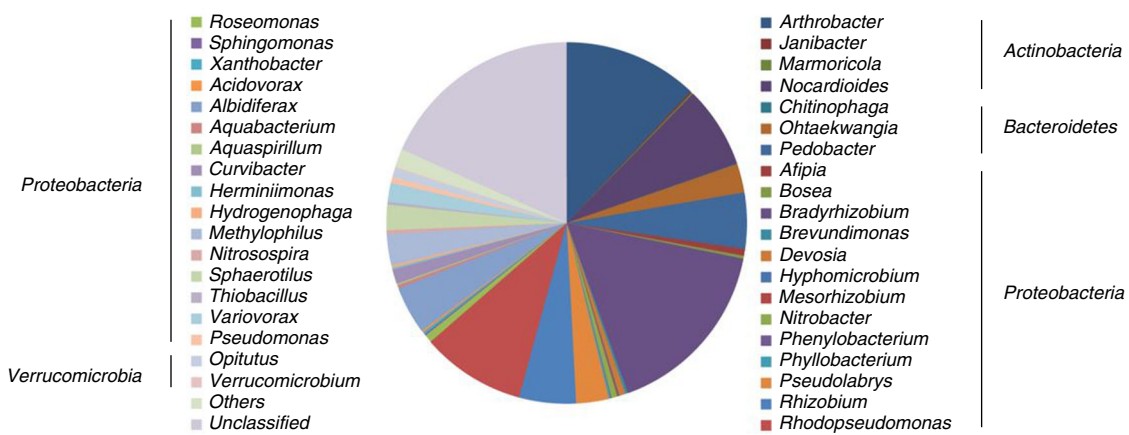

**Fig. 9** The composition of bacterial communities at genetic level. The genera contained <0.1% of sequence reads were summarized in group "others"

Actinobacteria, and *Pedobacter* (9.88%) in Bacteroidetes. *Bradyrhizobium* spp. are well known to be capable of fixing nitrogen and are important in nitrogen cycling[35]. Although primarily aerobic bacteria, *Arthrobacter* have been reported anaerobically reduce nitrate and ferment mixed acid[36], and have been isolated as the dominant lignin-degrading strains from brown coal[37]. *Pedobacter* spp. are known to be dominant in the anaerobic hydrolysis of cellulose to produce bio-hydrogen[38]. *Rhodopseudomonas* spp. were the dominant bacteria for hydrogen production in a microflora separated from anaerobic digestion sludge collected from a methane plant[39]. Most *Nocardioides* strains were isolated from habitats that contained aromatic compounds such as nitrophenol, phenanthrene and crude oil[40–42]. The other bacterial genera are also commonly found in other CBNG fields, such as *Acidovorax*, *Hydrogenophaga*, and *Rhizobium*. Two genera of sulfur reducing bacteria, *Desulfovibrio* and *Desulfosporosinus* were present in the inoculum with extreme low intensity (0.004%).

## Discussion

The inferences from the observed data in Fig. 5 are that six-carbon carbohydrates were metabolized more rapidly than the five-carbon carbohydrates. The six-carbon carbohydrates are generally metabolized through glycolysis[43]. Five-carbon carbohydrates are generally metabolized by anaerobic microorganisms through the pentose phosphate pathway, which requires several steps before entering glycolysis. Recent research comparing mixed cultures and the use of xylose found higher fractions of carbon converted into catabolic products (e.g., acetate) and the cellular biomass yield was 20% lower when compared to glucose as the substrate[44]. New research has found that fermenters of xylose, when compared to glucose, accumulate acetic acid[45]. This accumulation could offset the dynamics of the microbial community in a mixed culture, and result in rate differences. Since the biological metabolism is different between five- and six-carbon sugars, there could be an indirect impact towards the methane production, resulting in differences between the five- and six-carbon carbohydrate performances in an anaerobic coalbed microcosm. While the statistics conducted on the entire Gompertz modeling sets suggested keeping all treatments separate, biological means of metabolism were determined to be appropriate for grouping the five-carbon carbohydrates and the six-carbon carbohydrates. When comparing the linear rates, the results for xylose and arabinose were statistically similar while glucose, galactose, and mannose were also statistically similar.

Five-carbon sugars may be advantageous for enhanced methane production. While the five-carbon carbohydrates may not benefit the fermentative microorganisms for cellular production, the higher methanogenic precursor end products could provide more methanogenic substrates than glucose and other six-carbon carbohydrates. Other studies involving the fermentative bacterium *Paenibacillus polymyxa* reported similar occurrences[45], where more acetic acid, less acetate uptake, and higher instances of a more direct pathway for acetate production were observed on xylose grown cells when compared to glucose grown cells[45]. Although statistical differences in the five- and six-carbon carbohydrate methane production were not observed, differences may be observed, when grouped and averaged based on the biological pathways. However, other studies have shown observable trend due to the high variation of mixed culture experiments[44–46]. The standard deviation of the replicates samples tends to be more significant when gas production starts. The uncertainty diminished over time towards the end of incubation (Fig. 2), which may be associated with the very low microbial quantity in the coal ("Microbial community analysis" Section). Biomass accumulation is relatively slow, where differing initial growth rates effects when gas production is observed; therefore, larger error bars represent the variability of the initial biomass growth, but the variability decreases with biomass accumulation.

Cellobiose took longer to degrade in the microcosm incubation, due to its more complex structure, when compared to the monosaccharides tested. The unique and slightly recalcitrant β-1, 4 bond that links two glucose bonds of the cellobiose structure are less soluble and require unique enzymes to degrade the compound. While monosaccharides, in general, can be transported across the membrane of microbial cells directly, disaccharides, such as cellobiose, require enzymes from within the cells to be excreted to degrade the cellobiose compound first, and then can be transported across the membrane into the cell for metabolism[47]. As in the cellulosic biofuels industry, hydrolysis and consumption rates of the primary fermenters are slower on five-carbon carbohydrates and cellobiose[47, 48]. Secondary biogenic coalbed natural gas regeneration using biostimulation of abundant carbohydrates might expect less time for acclimation of the six-carbon carbohydrates than the five-carbon carbohydrates. Methane can be formed in a shorter time scale in the six-carbon carbohydrates than in the five-carbon carbohydrates.

Anaerobic zones tend to have low COD balances (<80%), where the fermentation is the major factor contributing toward the low COD balance[49]. Fermentation is a low energy process that forms high-energy byproducts, which can be further degraded or sometimes directly used by methanogens. The stoichiometric yields show that six-carbon carbohydrates can produce more methane than five-carbon carbohydrates (2.4 vs. 2) on a molar basis. However, methane production efficiencies on xylose ranged from 51 to 67% were observed[50]. Methane production on glucose with a consortium has been reported to higher (near 90%) with a maximum specific methane yield of 2.13 mol mol$^{-1}$ glucose which is close to theoretical value[51]. The lower conversion efficiencies in this study could have been affected by the non-acclimated consortia and possible inhibition by pH. The acetate control yielded an average of 0.4 mol mol$^{-1}$ acetate compared to the theoretical 0.8 mol mol$^{-1}$ acetate. The cellobiose substrate-fed system was drastically different. The conversion efficiency was only 2%, which was the lowest efficiency of the tested substrate. However, it has the highest specific yield of 5.8 moles methane per mole cellobiose. This is an indication that cellobiose was either not bioavailable in the system or was inhibited by external conditions.

Lower conversion efficiencies of coal microbial communities may be due to the contents and compounds within the coal itself, causing interferences with the microbial performance. In addition to the low conversion efficiencies, the correlation of the energy content study could both have been affected by abiotic processes such as diffusion and sorption of the coal system. Toxic compounds within coal could have affected the methanogenic yields, where Orem et al. (2010)[52] observed PAHs, phenols, biphenyls, and aromatic amines were present in coalbed water at concentrations between 18 and 100 µg l$^{-1}$. Evidence of coal-derived compounds from Wyodak coal have been detected in the study of intermediates. Various hydrocarbons were detected by Gas Chromatography-Mass Spectroscopy (GC-MS) and were low at the start of the tests and increased during methane production and then decreased after stationary phase of the methanogens. These organics included hexadecanoic acids, phenols, sitosterols, oleic acids, and other long chain fatty acids, alkanes, and low molecular weight aromatics[52]. These products such as phenols can be toxic to methanogenic processes and resulted in reducing production by as much as half[21]. More important than toxicity, the low conversion efficiencies could have been due to sulfate reduction out-competing the methanogens for electron donors. The concentration of sulfide and sulfate are 4.2–29.2 µg l$^{-1}$ and 1.7–3.3 mg l$^{-1}$, respectively. Microbial community analysis

showed that two genera of sulfur reducing bacteria, *Desulfovibrio* and *Desulfosporosinus* with low intensity (0.004%) were found in the coal sample. However, these suggest that the sulfur reducing bacteria may have minimal impact on the conversion efficiency. Other studies do not clearly account for the added carbon of the biological media used, such as yeast extract, tryptone broth, and the carbon sources are not evaluated for their yield and if the electron donors could have been utilized by other respiring microorganisms[53–55].

Methane from coal seams was once thought to have formed millions of years ago when the coal itself was being formed. However, recent scientific discoveries have shown that much of the natural gas can be attributed to subterraneous anaerobic microbial systems living within the coal seams. Since these microorganisms use coal as a substrate or food source, the secondary biogenic natural gas that they produce can be considered a carbon-positive source of unconventional natural gas. In contrast, plant-derived biogenic coalbed natural gas is an innovative way of producing a source of low carbon renewable natural gas (carbon neutral or carbon negative) because the carbon dioxide emitted during combustion previously resided in the atmosphere prior to being fixed into biomass through photosynthesis. The carbon intensity of the gas can be further reduced through the capture and storage of biomass and coals affinity to adsorb carbon dioxide (Carbon emission for delivery of carbon source to the coal seams accounts for only 2–5% of the total carbon injected, based on our calculations.). However, producing low carbon renewable natural gas from coals seams requires an understanding of the indigenous microbial community associated with these reservoirs. In this study the microorganisms in the coal were shown to anaerobically metabolize monosaccharides (glucose, galactose, mannose, arabinose, and xylose) to methane to a greater extent than the disaccharide (cellobiose). Another important finding of this study is that carbohydrate-fed incubations did not lower the heating value of coal.

This study provides base line information on laboratory yields, rates, intermediate products, and impacts on energy content and can add to future assessments toward the use of plant-derived carbohydrates as an alternative carbon source for enhancing the production of methane in PRB coal. Capitalizing on the most abundant bioavailable carbon sources on Earth may serve as effective and economical enhancement strategies for this low carbon renewable natural gas. Plant-derived biogenic coalbed natural gas is an innovative way to utilize the cycling of carbon that occurs in nature. With the wide spanning coal seam as the natural bioreactor, existing infrastructure, and an understanding of microbial mechanisms, this cleaner fuel has great potential for replacing burning coal for energy. In addition, carbon credits could significantly increase the value of this LCRNG (an additional value of $9.55 MMBtu$^{-1}$ compared to commodity price of natural gas of $3.25 MMBtu$^{-1}$, Prices as of 19 May 2017)[56]. In conclusion, this may have important ramifications in the generation of low carbon renewable natural gas from added carbon sources within geological formations and is an important approach in the development of more sustainable and cleaner alternatives to fossil fuels. With carbon neutral/carbon negative natural gas, it has implied and will fit well into the context for the collective global efforts of reducing carbon footprint to mitigate the global warming.

## Methods

**Coal sample and inoculum**. The coal and water samples were obtained from the Fort Union Formation at a site located approximately 60 miles south of Gillette, WY, within the PRB. The sampling well (FED 41–18) is located at coordinates 43.61909° North and 105.54648° West. The coal sample was collected from the same well with depths ranging between 312–321 meters below ground surface (bgs)

from Canyon coal in the Wyodak coal zone. The sample was vacuum sealed, and then stored under nitrogen gas ($N_2$) at 4 °C until used. The well had a cumulative approximate gas production of 5094 MCF ($1.4 \times 10^5$ m$^3$) and water production of 5716 barrels or 908,771 liters[57]. To view the basic lithology, described as the layers of rock and coal, see in Supplementary Fig. 2.

The coal is subbituminous in rank with a heating value of 11,440 Btu lb$^{-1}$ (26,609 kj kg$^{-1}$) on a moisture-free basis. The elemental composition of the coal is presented in Supplementary Table 3 (Wyoming Analytical Laboratories, Inc., Laramie, Wyoming, USA). Prior to the use of the coal as inoculum, it was ground in an anaerobic chamber using an autoclaved mortar and pestle, mixed and sieved to ensure consistency. The fraction of coal passing through a 60 mesh (250 µm) sieve was used as the inoculum for all of the bottles.

**Nutrient medium**. Stock solutions of minerals, trace metals, and vitamins were prepared as described in Hurst (2007)[58]. The contents and their concentrations in the mineral solution included: 0.8 g l$^{-1}$ NaCl, 1 g l$^{-1}$ NH$_4$Cl, 0.1 g l$^{-1}$ KH$_2$PO$_4$, 0.1 g l$^{-1}$ KCl, 1.62 g l$^{-1}$ MgCl$_2$·6H$_2$O, and 0.04 g l$^{-1}$ CaCl$_2$·2H$_2$O. The contents and their concentrations in the trace metal solution included: 20 mg l$^{-1}$ Nitrilotriacetic acid, 10 mg l$^{-1}$ MnSO$_4$·H$_2$O, 8 mg l$^{-1}$ Fe(NH$_4$)$_2$(SO$_4$)$_2$·6H$_2$O, 2 mg l$^{-1}$ CoCl$_2$·6H$_2$O, 2 mg l$^{-1}$ ZnSO$_4$·7H$_2$O, 0.2 mg l$^{-1}$ CuCl$_2$·2H$_2$O, 0.2 mg l$^{-1}$ NiCl$_2$·6H$_2$O, 0.2 mg l$^{-1}$ Na$_2$MoO$_4$·2H$_2$O, 0.2 mg l$^{-1}$ Na$_2$SeO$_4$, and 0.2 mg l$^{-1}$ NaWO$_4$. The contents and their concentrations in the vitamin solution included: 0.1 mg l$^{-1}$ pyridoxine HCl, 0.05 mg l$^{-1}$ thiamine HCl, 0.05 mg l$^{-1}$ riboflavin, 0.05 mg l$^{-1}$ calcium pantothenate, 0.05 mg l$^{-1}$ thioctic acid, 0.05 mg l$^{-1}$ p-aminobenzoic acid, 0.05 mg l$^{-1}$ nicotinic acid, 0.05 mg l$^{-1}$ vitamin B$_{12}$, 0.05 mg l$^{-1}$ MESA mercaptoethanesulfonic acid, 0.02 mg l$^{-1}$ biotin, and 0.02 mg l$^{-1}$ folic acid. The only difference in the recipe used in this study was the substitution of 0.083 g l$^{-1}$ MgCl$_2$·6H$_2$O for MgSO$_4$·7H$_2$O (0.01 g l$^{-1}$) to reduce the amount of sulfate added, in an effort to reduce the electron loss to sulfate reducing bacteria. Sodium bicarbonate with a final concentration of 1 mg l$^{-1}$ was added to buffer the solution.

The vitamin solution was not added until the anaerobic chamber. Within the anaerobic chamber, the vitamin solution was added and 0.1 g l$^{-1}$ sodium sulfide nonahydrate, which provided a reducing environment (Eh below −300 mV), as redox couples (H$_2$/H$^+$ and CO$_2$/CH$_4$) are approximately −400 to 200 mV. 1 mg l$^{-1}$ resazurin in the final solution was an indicator to monitor for a change in redox conditions and is colorless below −110 mV and turns pink under conditions above −50 mV[25]. General conditions for anaerobic culturing were attained at low electro-potential, optimal at a temperature of 30°C and a pH range of 5.9–8.8[25].

**Anaerobic microcosms**. There were a total of seven coal microcosm treatments prepared in triplicate with three controls (Supplementary Table 4). The study had three replicates of each treatment (glucose, galactose, mannose, arabinose, xylose, cellobiose, mix, the negative control, and two positive controls). Each individual consortium culture was homogeneous with respect to coal type, location, size, and coal composition. Two positive controls were established from the two main substrate pathways for methanogens. The acetic acid positive control was established for detecting the acetoclastic methanogens while the positive hydrogen control was established for detecting the hydrogentrophic methanogens that can use hydrogen and carbon dioxide as substrates. The negative control contained the coal inoculum, nutrient media, buffer, redox indicator, and reducing compound but no alternative carbon source other than the coal inoculum. The incubations took place in the dark at a temperature of 30 °C.

The microcosm study tested the ability of the coal-derived microorganism to degrade a wide range of plant-derived carbohydrates including hexose carbohydrates D-glucose (Spectrum), D-mannose (Sigma), D-galactose (Difco), pentose carbohydrates D-xylose (99+%, Acros Organics) and L-arabinose (Acros Organics), and a more complex carbohydrate, D-cellobiose (Fluka). The target concentration of the substrates was 10 mM as suggested in Hurst (2007)[58]. Having the same amount of carbon in each treatment was important, and the 10 mM concentration allowed for approximately the same weight ratios of hydrogen, carbon, and oxygen, even though the structures and weights of the compounds were different.

**Gas analysis**. The gases of interest, including methane, hydrogen, carbon dioxide, and oxygen, were analyzed using a PerkinElmer Clarus 500 gas chromatograph (GC) equipped with a Carboxen 1010 porous-layer open tubular column (PLOT) with dimensions (30 m × 0.32 mm). Detail components of the GC include a thermal conductivity detector ran at 225 °C and the injector ran at 200 °C. The carrier gas was ultra-high purity nitrogen with a flow rate of 2 ml min$^{-1}$ and a split ratio of 8:1. The oven was programmed to start at 40 °C for 1 min, and then increased at the rate of 20 °C min$^{-1}$ to 120 °C for 2 min. The headspace gas composition of the microcosms was measured at a set interval. The bottles were tested for gas composition with a sample size for the gas analysis of 30 µl, withdrawn using a 50 µL SampleLockTM gas-tight syringe (Hamilton, Reno, NV) with a 26-gauge side-hole removable needle (Hamilton, Reno, NV).

**Substrate and organic analysis**. Monitoring of carbohydrates and organic acids was conducted using an Agilent high-performance liquid chromatography (HPLC) equipped with an Aminex HPX-87H cation-exchange column (300 by 7.8 mm)

specifically for organic acids and simple carbohydrates (no. 125-0140; Bio-Rad laboratories, Hercules, CA), a micro-guard column (no. 125-0129: Bio-Rad), and a holder (no. 125-0131; Bio-Rad). The signal settings were: UV wavelength of 210 nm with band width 10, reference 360 with band width 100. The spectrum settings had a step of 2, range of 190 through 600 nm. The physical settings were: flow was 0.6 ml min$^{-1}$, with a mobile phase of 0.0005 M $H_2SO_4$. The injection amount was 30 µl. A 20 µl bypass sampler loop was used and the UV Absorber was set to 210 nm. The band width was 8 nm. The HPLC was set up with two detectors in tandem, one to detect the carbohydrates, the refractive index detector (RID) and the other to detect the organic acids, the diode array detector (DAD). The RID used was a Waters 401 refractometer, for identification and quantification of carbohydrate content. The following settings were used for the Waters 401 refractometer: sensitivity 32, and temperature 50 °C. The DAD used was a G1315B Diode Array Detector. Chemstation software was set up with the equipment to perform and organize the calibrations, integration calculations, and quantification of the major carbohydrates and organic acids. pH was also measured using a Mettler Toledo S220 pH meter (Columbus, OH, USA).

**Energy content analysis**. At the end of the incubation period the coal from the microcosm bottles was tested to evaluate the effect that microbial action had on the energy content of the coal. Four dry mass calorific value measurements of coal from the five-carbon, six-carbon, incubated control, and non-incubated control experiments were analyzed. Each of the treatments were independent experimental units, because they did not depend on each other. All samples were derived from the same well, were crushed to approximately the same size, and were assumed to be homogenous. One-way ANOVA, coupled with a post test of Least Significant Difference (LSD) was performed to determine whether there was a statistically significant difference between the means t and a statistical package, SAS statistical software, was used to compare the results.

**Modeling**. A modified Gompertz sigmoidal curve Eq. (2) was used to model methane production[59, 60].

$$y = A * \exp\left[-\exp\left[\left(\frac{\mu_\mathrm{m} * \exp(1)}{A} * (\lambda - t) + 1\right)\right]\right] \tag{2}$$

where $y$ is the methane yield; $A$ is the methane potential; $\mu_\mathrm{m}$ is the growth rate; $\lambda$ is the lag time and $t$ is the incubation time. The model was used to estimate the cumulative rates of methane formation of the five-carbon, six-carbon, and cellobiose treatments. The Marquardt algorithm for non-linear regression was used to fit the model to the data[61, 62]. The algorithm estimates values of model parameters by minimizing the sum of the squared differences between the observed and the predicted values. The final cumulative methane values after the 128-day incubation and the lag times were analyzed using the Gompertz model while growth rate was analyzed using the Monod equation. The purpose of the modeling was to provide quantitative comparison of the magnitude and rate of methane production associated with arabinose, xylose, galactose, mannose, glucose, cellobiose, and the mix of carbohydrates. The residuals were determined normal by plotting them and finding a common cylindrical shape around day 30 to day 106 (data not shown). Variances were determined using a Hartley table threshold for the individual treatment amounts, replicate number for the various treatments and controls, and a 0.05 alpha, which is the maximum willingness to make a type I error of incorrectly accepting the null hypothesis.

On the basis of stoichiometry of cell synthesis reactions for methanogenesis, Rittman and McCarty (2001)[21] developed the following general reaction (3) assuming $CO_2$ as the electron

$$C_nH_aO_bN_c + \left(2n + c - b - \frac{9df_s}{20} - \frac{df_e}{4}\right)H_2O \rightarrow$$
$$\frac{df_e}{8}CH_4 + \left(n - c - \frac{df_s}{5} - \frac{df_e}{8}\right)CO_2 + \frac{df_s}{20}C_5H_7O_2N\left(c - \frac{df_s}{20}\right)NH_4^+ \left(c - \frac{df_s}{20}\right)HCO_3^- \tag{3}$$

where,

$$d = 4n + a - 2b - 3c$$

The reaction (3) accounts for cell biomass with $f_s$ and $f_e$, as well as the flow of electron donor substrate as an input, and the products and cell residual as the output. One portion of the electron donor goes towards reaction end products and energy production ($f_e$), and the other toward the synthesis of active bacterial cells ($f_s$): the sum of $f_s + f_e = 1$[21]. This approach served to track the detailed chemistry and thermodynamics of the microbial methanogenic reactions.

**DNA extraction and Miseq sequencing and analysis**. Coal was first pulverized and washed using a 0.05 M sterile phosphate buffer (pH 7.4) containing 0.2% Tween 80 as described previously[33] to elute the microorganisms attached to the surface of coal and eliminate interference of heavy metals which might reduce the efficient of DNA extraction. After washing, the solutions were filtered through a

0.22-µm membrane filter (Whatman, Japan). The filter was used for DNA extraction with a FastDNA SPIN kit for soil (Bio101 Systems, Carlsbad, USA) according to the manufacturer's instructions. DNA quality and concentration was determined using a Nanodrop™ 1000 Spectrophotometer (Nanodrop, Wilmington, DE, USA) and agarose gel electrophoresis. The obtained DNA was stored at −20 °C for further processing.

DNA sequencing was performed by the Institute for Environmental Genomics at the University of Oklahoma with an Illumina Miseq platform (San Diego, CA, USA). The protocol of PCR, sequencing and data analysis were performed as described previously[63]. Briefly, the 16S rRNA variable region 4 (V4) were amplified using the universal primer sets 515F/806R (GTGCCAGCMGCCGCGGTAA/GGACTACHVGGGTWTCTAAT) which have been used for Illumina-based surveys[64] to amplified bacterial and archaeal 16S rRNA genes[65]. The raw sequence reads were processed by an in-house sequence analysis pipeline (IEG sequence analysis pipeline, http://zhoulab5.rccc.ou.edu). Btrim was used to complete the quality trimming with minimum length of 150 bp, average quality score of 25, and removing sequencing adaptors[66]. Forward and reverse reads were combined using FLASH. Chimeric sequences were removed using UCHIME. The operational taxonomic units (OTUs) were classified using UCLUST[67] at the 97% similarity level, and singletons were removed. The taxonomic assignment was conducted by RDP classifier[68] with minimal 50% confidence estimates. The 16S rRNA gene sequences derived from Miseq pyrosequencing have been deposited in the NCBI Sequence Read Archive with the accession number SRP094828.

**Data availability**. The 16S rRNA gene sequences derived from Miseq pyrosequencing have been deposited in the NCBI Sequence Read Archive with the accession number SRP094828. The rest of the data supporting the findings of this study are available within the article and its Supplementary Information files. Extra data are available from the corresponding author upon request.

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

## Acknowledgements

Funding for this project was provided by RPSEA through the "Ultra-Deepwater and Unconventional Natural Gas and Other Petroleum Resources" program authorized by the US Energy Policy Act of 2005. The authors would like to thank Bob Vergnani (Coleman Oil and Gas, Inc.) for collecting the coal and water samples used in this study. The authors also thank Dr. Vikram Chhatre of INBRE Bioinformatics Core at the University of Wyoming for helping on phylogenetic tree plotting and Dr. Liam Revell of Department of Biology at the University of Massachusetts for providing phylogenetic tree file converting tool.

## Author contributions

Z.H., C.S., M.A.U., H.G., Q.W. conceived the idea of the experiment and were responsible for study design, interpretation of results and manuscript writing. All of the authors reviewed, approved and contributed to the final version of the manuscript.

## Additional information

**Competing interests:** The authors declare no competing financial interests.

