## [Peer Review File · Nature Communications]

Reviewers' Comments:

Reviewer #1 (Remarks to the Author)

Reviewer: Tim A Moore (I am fine with being identified as the reviewer)

"Renewable coalbed natural gas using plant-derived carbohydrates as an alternative carbon ... "
Huang et al.

I would recommend that this paper can be published, but not as it is; at least moderate revision is needed. See below.

A. This paper is mostly about organic chemistry and I am not an organic chemist. However, there are some aspects of geology that I can comment on, although these will be limited.

B. The overall English of the paper is good, though there are some awkward sentences and outright misuse of words (e.g. line 36 it sure be 'natural gas' not 'nature gas'). I'll leave the authors to search out correct these types of mistakes

C. The biggest issue I had with the paper was that it did not make clear a few points. These are listed below:

a. Are the authors suggesting that biomass is injected into the coal seam or some product of biomass? Surely not as that is fairly impossible. I don't understand the statement of using coal seams as a "bioreactor".

b. If the authors are saying that they are using the microbes from the PRB seams to digest biomass at the surface, then this has to be made clear from the start. As it is written this is not clear at all.

c. If (b) above is true, I'm not sure why the authors used coal samples to test for methanogenesis; something just isn't clear here ...

d. If (a) above is true, then should they not be considering the injection scenario in this? Coal material swells at the drop of a hat! All other kinds of things happen geochemically as well and its low permeability to start out with. If the authors are truly envisioning using the actual coal seams (buried) as some sort of bioreactor, how? How could this feasibly be done? Maybe I'm missing something here, but this needs to be clarified.

D. It has been shown by some other authors that the organic composition of the coal affects significantly the level of biogenic gas generation. Should not the authors have characterized the samples as their organic composition?

E. Did the authors isolate, characterize and type the microbial community in the samples? Are they sure that the inoculum was exactly the same in all samples and experiments?

F. The article may be very interesting for organic chemists, but I think a wider and fundamentally more interested audience, would be achieved if the article also addressed:

a. The geological environment and constraints on any 'bioreactor' and what exactly that means

b. What is the microbial fauna? How were these defined if they were defined. Should they not have been defined? If no why not?

c. Other than showing that PRB microbes can degrade certain organic material, how to get from that to a 'renewable' energy source? That is a big jump and the authors don't justice to their work by not leading the readers on as to how this might be possible.

G. At the end of the paper the authors note that they will be doing a pilot in the Hanoi basin. Once again I'm confused as what and how will such a field demonstration be conducted. More detail is needed, not a lot of detail, but enough so the audience can actually understand what it is the paper is ultimately proposing.

As a footnote, I worked in the Hanoi Basin doing CBM exploration drilling in 2009 and 2010. We

found very little to no biogenic gas. Thus I'd be interested in knowing if there are microbes there – it is a very hydrodynamic basin and even at 1000 m there were relatively fresh water aquifers. A possible explanation as why there were no microbes ... possibly. Anyway, I make those comments line 709 in the authors manuscript states they might be working on a pilot in the Hanoi basin.

Reviewer #2 (Remarks to the Author)

Remarks to the author

In the manuscript entitled "Renewable coalbed natural gas using plant-derived carbohydrates as an alternative carbon source and the implication of carbon capture and storage", Huang et al. report the use of monosaccharides and disaccharides to increase methane production in microbial cultures indigenous to the PRB coal seam. The authors carried out anaerobic microcosm experiments that were chemically and model based analysed to support their claim that an indigenous microbial community from a coal seam is capable to convert plant-derived carbohydrates to methane.

This is an interesting study and the overall goal to use depleted or non active coal seam parts as in situ anaerobic digester for methane and energy generation could be of interest for the coal mine enterprises.

In the last years several studies have been carried out to show that methane can be generated as biodegradation end product of plant-derived biomass feedstock including sugars as monosaccharides and disaccharides. It is also known that coal indigenous microbial communities are usual comprised of fermentative and acetogenic bacteria as well as methanogens able to utilize parts of coal as carbon source or amended substrates as acetate and complex organic matter to enhance methane production in situ.

To the best of my knowledge, the use of different types of monosaccharides and disaccharides to feed to an indigenous coal community for methane production hasn't been reported yet. However, I question the weighting of these experiments for the in-situ application and the overall aim. I think the report of the actual field trial would be without doubt suitable for a publication in the journal nature communications in addition to the authors idea already patented in 2012 (Biomass-enhanced natural gas from coal formations, WO 2012135847 A1). In my opinion the observed experiments are better suited in a biochemistry journal so far.

Of major concerns are the conditions the microcosms were carried out that could be more accurate to actual in situ conditions. Additionally, the failure of the buffering system resulting in a huge decrease of pH in the cultures questions the suitability of these incubations as preliminary experiment for the field trial. Furthermore, I would strongly suggest a more comprehensive microbiological analyses of the experiments in order to close the "suggestion gaps" and gain a broader picture on the actual biodegradation processes. Since the indigenous microbial community is a key parameter in that experiment for the turnover of biomass, it would be of immense interest what the community looks like.

The lab incubations are carried out using anaerobic media. Why not using coal seam water to keep it as accurate as possible to the in situ conditions? Is the plan to add vitamins and trace elements to the coal seam later (high costs)? The pH is an important parameter for the success and can be quite different in situ compared to your controlled lab experiments, do you account for that? How important are the microcosms still for the pilot study of the field trial since the pH changed a lot (Line 570-571)?

There was no addition of sulfate to the medium in order to enhance methane production but what about the sulfate concentrations in the coal seam?

Line 164. Resazurin is a nice electron shuttle and can alter microbial communities.

Lines 345-352 and 506-508. Speculations could be resolved and case strengthen by community

analyses (e.g. Illumina sequencing) and quantitative PCR of 16S rRNA and functional genes, e.g. for methanogens and sulfate reducers. Additionally, hints could be gained for detected lag phases in the incubations.

Minor concerns

Title. Change coalbed natural gas to natural gas as coal hasn't actually shown an impact on the methane generation. The gas is produced through the addition of carbohydrates.

Line 78. Plural show not shows.

Line 322: Needs to be more clear. Did you add an electron acceptor (CO₂) to the hydrogen positive control?

Line 535: H₂S and SO₄²⁻ concentrations could be easily analysed to link with the relative abundance of sulfate reducers. What is the origin of H₂S if no sulfate was added?

General. Some repetitions in the text that could be avoided.

Reviewer #3 (Remarks to the Author)

A. Summary = plant-derived biogenic coalbed natural gas could yield a carbon neutral or negative energy process.

B. Originality = the concept does appear to be novel although others have looked at stimulation of biogenic systems such as coal or heavy oil etc e.g. see IM Head, DM Jones, SR Larter Nature 426 (6964), 344-352, S Larter, M Strous, S Bryant

EGU General Assembly Conference Abstracts 18, 17380, IMC Head, SR Larter, ID Gates

US Patent 9,187,687, G Sentharamaikkannan, K Budwill, I Gates, S Mitra, V Prasad

Energy & Fuels 30 (2), 871-883

Interest = from academic side, yes, for zero or negative carbon energy processes, from commercial side - not sure if much interest since gas prices low and current trend is towards hydraulic fracturing tight gas resources

C. Validity of approach = approach is reasonable and complete. Quality of data = data shows severe scatter e.g. Fig 4, 5, 6, and model fits are within a cloud of data e.g. Fig 6. This suggests uncertainty in the results of the experiments. No error bars are displayed - I would suggest that the results are re-examined with error margins stated with uncertainty analysis done on the model fits and how this impacts the overall conclusions of the research. Presentation is clear. One thing that I did not see was the energy required for the overall process such as pump energy to inject stimulants into the coalbed - this should be accounted for taking the physical system underground into account (it is not often high permeability as is the media in the experiments) - in the coal seam, the permeability can be very low and the energy required to move injectants within the seam can be high (and offset the carbon neutrality of the process). The key missing link between these kinds of experiments where mass transfer is not a limitation and the actual coal seam (and its issues of mass transfer = need to often fracture deep coal to get gas out of it at commercial rates) is that the experiments eliminate mass transfer limitations which is often the crucial limitation for in situ processes.

D. Stats = appropriate but as stated for C., the data has high variability and I'd suggest add error measures to plots.

E. Conclusions = conclusions make sense given the results of analysis but again, see C., I'd suggest evaluation of uncertainties and impact on conclusions.

F. Improvements = given data, after uncertainty analysis, I'd suggest potentially repeating the experiments to bracket the uncertainty range.

G. References = suggest add more references specifically on kinetics of biodegradation of coal since this is a critical part of the work and compare earlier results to kinetics from experiments reported to see if similar order of magnitude etc. This will provide an additional step towards validity.

H. Clarity = there are a few typos and grammar errors but manuscript is clear, abstract is appropriate.

Reviewers' comments:

Reviewer #1 (Remarks to the Author):

Reviewer: Tim A Moore (I am fine with being identified as the reviewer)

“Renewable coalbed natural gas using plant-derived carbonhydrates as an alternative carbon ...” Huang et al.

I would recommend that this paper can be published, but not as it is; at least moderate revision is needed. See below.

A. This paper is mostly about organic chemistry and I am not an organic chemist. However, there are some aspects of geology that I can comment on, although these will be limited.

B. The overall English of the paper is good, though there are some awkward sentences and out right misuse of words (e.g. line 36 it sure be ‘natural gas’ not ‘nature gas’). I’ll leave the authors to search out correct these types of mistakes

Response: The misuse and typo were corrected.

C. The biggest issue I had with the paper was that it did not make clear a few points. These are listed below:

a. Are the authors suggesting that biomass is injected into the coal seam or some product of biomass? Surely not as that is fairly impossible. I don’t understand the statement of using coal seams as a “bioreactor”.

Response: The derivatives of the biomass such as simple carbohydrates/sugars/waste from sugar process facilities contain soluble sucrose are proposed to be injected into the coal seam. This was clarified in the revision in the Introduction. A clearer account of using coal seams as geobioreactors was also provided in the same section.

b. If the authors are saying that they are using the microbes from the PRB seams to digest biomass at the surface, then this has to be made clear from the start. As it is written this is not clear at all.

Response: The indigenous microbes will not be used to digest biomass at the subsurface. The indigenous microbes were used to evaluate the potential of gas production with carbon source external to coalbeds. This was further clarified in the Introduction section. The sugars/carbohydrates/waste and other soluble components which are derivatives

from biomass at the surface facilities. All potential method such as fermentation, oxidation etc. will be considered.

c. If (b) above is true, I'm not sure why the authors used coal samples to test for methanogenesis; something just isn't clear here ...

Response: (b) is not true as what authors proposed is to inject soluble components such as sugars derived from biomass at the surface facilities. The derivatization can be by any means that are sustainable and cost effective. The reason of using coal samples as inocula is to test ability of gas production by microbial communities in coal. Typically, in the biogenic coalbed natural gas research, people prefer to use native microorganisms over non-native as this would overcome some of the hurdles exerted by regulation when apply such technologies in the field. We made this clear in the Introduction.

d. If (a) above is true, then should they not be considering the injection scenario in this? Coal material swells at the drop of a hat! All other kinds of things happen geochemically as well and its low permeability to start out with. If the authors are truly envisioning using the actual coal seams (buried) as some sort of bioreactor, how? How could this feasibly be done? Maybe I'm missing something here, but this needs to be clarified.

Response: (a) is not true as the sugars solution rather than biomass (solid) will be injected into the coal seam. There are micropores, mesopores and macropores in the coal. The macropores basically consist of cleats with size of milimeters. Lower rank coal typically has higher permeability. The mass transfer may be easily overcome in coal seams with high permeability such as Powder River Basin.

D. It has been shown by some other authors that the organic composition of the coal affects significantly the level of biogenic gas generation. Should not the authors have characterized the samples as their organic composition?

Response: The proximate data including volatile matter and fixed carbon were added in the Table 1 along with the ultimate analysis. Our studies showed (Huang et al., 2013 b in the reference) that without any pretreatment by chemicals or enzymes, the coal will not produce any gas using indigenous microbes. That is to say, the organic composition will have no impact on the gas production. This is consistent with the coal heating value tests, as the heating values were not reduced after incubation (carbon were not transformed).

E. Did the authors isolate, characterize and type the microbial community in the samples? Are they sure that the inoculum was exactly the same in all samples and experiments?

Response: The coal was ground using a mortar and pestle and, then sieved and well mixed, in an anaerobic chamber before inoculation. The same ground sample was used to inoculate for all the microcosms. This was to ensure that the inoculum was consistent. A

description regarding this matter was added in the methods section. A microbial analysis of the inoculum was added.

F. The article may be very interesting for organic chemists, but I think a wider and fundamentally more interested audience, would be achieved if the article also addressed:

a. The geological environment and constraints on any 'bioreactor' and what exactly that means

Response: A brief discussion of the geological environment and constraints was added.

b. What is the microbial fauna? How were these defined if they were defined. Should they not have been defined? If no why not?

Response: A microbial composition analysis of the inoculum was done and added.

c. Other than showing that PRB microbes can degrade certain organic material, how to get from that to a 'renewable' energy source? That is a big jump and the authors don't justice to their work by not leading the readers on as to how this might be possible.

Response: This was address in the Introduction section, with details.

G. At the end of the paper the authors note that they will be doing a pilot in the Hanoi basin. Once again I'm confused as what and how will such a field demonstration be conducted. More detail is needed, not a lot of detail, but enough so the audience can actually understand what it is the paper is ultimately proposing.

Response: A brief description of the project was added at the end of conclusion.

As a footnote, I worked in the Hanoi Basin doing CBM exploration drilling in 2009 and 2010. We found very little to no biogenic gas. Thus I'd be interested in knowing if there are microbes there – it is a very hydrodynamic basin and even at 1000 m there were relatively fresh water aquifers. A possible explanation as why there were no microbes ... possibly. Anyway, I make those comments line 709 in the authors manuscript states they might be working on a pilot in the Hanoi basis.

Reviewer #2 (Remarks to the Author):

Remarks to the author

In the manuscript entitled "Renewable coalbed natural gas using plant-derived carbohydrates as an alternative carbon source and the implication of carbon capture and

storage”, Huang et al. report the use of monosaccharides and disaccharides to increase methane production in microbial cultures indigenous to the PRB coal seam. The authors carried out anaerobic microcosm experiments that were chemically and model based analysed to support their claim that an indigenous microbial community from a coal seam is capable to convert plant-derived carbohydrates to methane.

This is an interesting study and the overall goal to use depleted or non active coal seam parts as in situ anaerobic digester for methane and energy generation could be of interest for the coal mine enterprises.

In the last years several studies have been carried out to show that methane can be generated as biodegradation end product of plant-derived biomass feedstock including sugars as monosaccharides and disaccharides. It is also known that coal indigenous microbial communities are usual comprised of fermentative and acetogenic bacteria as well as methanogens able to utilize parts of coal as carbon source or amended substrates as acetate and complex organic matter to enhance methane production in situ. To the best of my knowledge, the use of different types of monosaccharides and disaccharides to feed to an indigenous coal community for methane production hasn't been reported yet. However, I question the weighting of these experiments for the in-situ application and the overall aim. I think the report of the actual field trial would be without doubt suitable for a publication in the journal nature communications in addition to the authors idea already patented in 2012 (Biomass-enhanced natural gas from coal formations, WO 2012135847 A1). In my opinion the observed experiments are better suited in a biochemistry journal so far.

Of major concerns are the conditions the microcosms were carried out that could be more accurate to actual in situ conditions. Additionally, the failure of the buffering system resulting in a huge decrease of pH in the cultures questions the suitability of these incubations as preliminary experiment for the field trial. Furthermore, I would strongly suggest a more comprehensive microbiological analyses of the experiments in order to close the “suggestion gaps” and gain a broader picture on the actual biodegradation processes. Since the indigenous microbial community is a key parameter in that experiment for the turnover of biomass, it would be of immense interest what the community looks like.

Response: A microbial composition analysis of the inoculum was done and added to fill the gap. A detail discussion was added in Section 3.6.

The lab incubations are carried out using anaerobic media. Why not using coal seam water to keep it as accurate as possible to the in situ conditions? Is the plan to add vitamins and trace elements to the coal seam later (high costs)? The pH is an important parameter for the success and can be quiet different in situ compared to your controlled lab experiments, do you account for that? How important are the microcosms still for the pilot study of the field trial since the pH changed a lot (Line 570-571)?

Response: We would like to keep this study as simple as we can so that we can focus on fewer variables and make a strong case to generalize the idea. Water is different for a coal seam to another, ranging from fresh (noted by the first reviewer in Hanoi Basin) to brackish. The purpose of supplementation of vitamin and trace elements was to ensure the gas production were not limited by nutrients. In a preliminary study we used MDS (molasses desugared solubles), a potential sugar containing waste for injection, greater amount of coal inoculum was added without nutrients supplementation. The results suggested that it offered enough buffer capacity and nutrient to produce comparable amount of methane. Coal has great pH buffer capacity and we believe pH will not cause any problem in field. Besides, in the field trial which is proposed to be a “plug flow reactor”, the dynamics of water will prevent proton of being accumulated.

There was no addition of sulfate to the medium in order to enhance methane production but what about the sulfate concentrations in the coal seam?

Coalbed waters are relatively low in sulfate because the chemical conditions in coal beds favor the conversion of sulfate to sulfide. Characterization is important to find those coal seams that meet criteria for the technology to be successfully implemented.

Line 164. Resazurin is a nice electron shuttle and can alter microbial communities.

Response: There is tradeoff of using resazurin; however, it is a good indicator of reducing condition. It is widely used in the research of this field. The alteration of microbial communities is out of the scope of this research.

Lines 345-352 and 506-508. Speculations could be resolved and case strengthen by community analyses (e.g. Illumina sequencing) and quantitative PCR of 16SrRNA and functional genes, e.g. for methanogens and sulfate reducers. Additionally, hints could be gained for detected lag phases in the incubations.

Response: A microbial composition analysis of the inoculum was done and added.

Minor concerns

Title. Change coalbed natural gas to natural gas as coal hasn't actually a shown impact on the methane generation. The gas is produced through the addition of carbohydrates.

Response: The term was changed as “low carbon renewable natural gas” to avoid the ambiguity.

Line 78. Plural show not shows.

Response: Corrected. Used “show” instead of “shows”.

Line 322: Needs to be more clear. Did you add an electron acceptor (CO₂) to the hydrogen positive control?

Response: It was already indicated in Table 2 that electron acceptor was added in the H₂ positive control. This is reiterated in line 310.

Line 535: H₂S and SO₄²⁻ concentrations could be easily analysed to link with the relative abundance of sulfate reducers. What is the origin of H₂S if no sulfate was added?

Response: PRB coal is well known for its low ash and sulfur content. However, there is still sulfur mostly in organic form in coal that can be transformed to other species. Two genera of sulfur reducing bacteria, *Desulfovibrio* and *Desulfosporosinus* were present in the inoculum with extreme low intensity (0.004%).

General. Some repetitions in the text that could be avoided.

Response: Repetitions were removed.

Reviewer #3 (Remarks to the Author):

A. Summary = plant-derived biogenic coalbed natural gas could yield a carbon neutral or negative energy process.

B. Originality = the concept does appear to be novel although others have looked at stimulation of biogenic systems such as coal or heavy oil etc e.g see IM Head, DM Jones, SR Larter Nature 426 (6964), 344-352, S Larter, M Strous, S Bryant
EGU General Assembly Conference Abstracts 18, 17380, IMC Head, SR Larter, ID Gates

US Patent 9,187,687, G Sentharamaikkannan, K Budwill, I Gates, S Mitra, V Prasad
Energy & Fuels 30 (2), 871-883

Interest = from academic side, yes, for zero or negative carbon energy processes, from commercial side - not sure if much interest since gas prices low and current trend is towards hydraulic fracturing tight gas resources

Response: The US EIA projected that the price of natural gas will be as much as four times of current price by 2030. More importantly, carbon neutral/carbon negative is the selling point. Economic analyses have shown that natural gas with carbon capture and storage is profitable in places like California which has stringent regulations greener energy policies.

C. Validity of approach = approach is reasonable and complete. Quality of data = data

shows severe scatter e.g. Fig 4, 5, 6, and model fits are within a cloud of data e.g. Fig 6. This suggests uncertainty in the results of the experiments. No error bars are displayed - I would suggest that the results are re-examined with error margins stated with uncertainty analysis done on the model fits and how this impacts the overall conclusions of the research. Presentation is clear. One thing that I did not see was the energy required for the overall process such as pump energy to inject stimulants into the coalbed - this should be accounted for taking the physical system underground into account (it is not often high permeability as is the media in the experiments) - in the coal seam, the permeability can be very low and the energy required to move injectants within the seam can be high (and offset the carbon neutrality of the process). The key missing link between these kinds of experiments where mass transfer is not a limitation and the actual coal seam (and its issues of mass transfer = need to often fracture deep coal to get gas out of it at commercial rates) is that the experiments eliminate mass transfer limitations which is often the crucial limitation for in situ processes.

Response: The error bars are added in Figure 4. An account of on the uncertain were added in Section 3.1. As explained in 3.1, The standard deviation of the replicates samples tends to be more significant when they start to produce gas. The uncertainty diminishes over time towards the end of incubation. This is typical for samples with extreme low microbial population. It takes time for biomass to accumulate while the accumulation of biomass to commence gas production for individual sample may slightly different. Hence, it shows greater error bars. Once the threshold is achieved, the uncertainty is reduced. Factors such as mass transfer, permeability, pump energy etc. are not included in this manuscript. They are essential for the actual injection. However, this will be done and evaluated in the subsequent pilot study.

D. Stats = appropriate but as stated for C., the data has high variability and I'd suggest add error measures to plots.

Response: The error bars were added to the plot.

E. Conclusions = conclusions make sense given the results of analysis but again, see C., I'd suggest evaluation of uncertainties and impact on conclusions.

The conclusion has not been changed

F. Improvements = given data, after uncertainty analysis, I'd suggest potentially repeating the experiments to bracket the uncertainty range.

Response: Uncertainty ranges were added in Table 3.

G. References = suggest add more references specifically on kinetics of biodegradation of coal since this is a critical part of the work and compare earlier results to kinetics from

experiments reported to see if similar order of magnitude etc. This will provide an additional step towards validity.

Response: To the best of our knowledge, none of research has been published and hence it is extremely difficult to find any reference on kinetics like this.

H. Clarity = there are a few typos and grammar errors but manuscript is clear, abstract is appropriate.

Response: Grammar errors and typos were corrected

Reviewers' Comments:

Reviewer #1 (Remarks to the Author)

I am fine with the changes made by the authors. It is indeed an interesting paper, though with the caveat from me that I can not evaluate the organic chemistry side of things. My only additional comments on the paper are:

1. Although a stratigraphic section has been given showing the stratigraphy of the sampling area, this hasn't been properly tied into the text. A reference is made in the text to the 'Wyodak' coal zone with the name 'Canyon' given. In fact, the Wyodak coal zone is both the Anderson and Canyon coal seams and their equivalents. Thus in figure 3 the word Wyodak needs to be properly placed for consistency with text. Also, a reference should be given in the text and figure caption for this (the authors might consider the reference (Flores, 2014, Coal and Coalbed Gas: Fueling the Future. Elsevier, Amsterdam, 697 pp. - it is a good place for a lot of the geology which is applicable to their sampling area).

2. The only response to one of my comments which I am still not fully happy with is their assertion that the organic composition of the coal won't impact on microbial activity. I disagree strongly on that and although not assessed in the experiments discussed in this paper there should at least be some discussion on this topic. The authors should see section 3.1 of Moore (2012) and especially read the thesis of Mares (2009) and the paper by Papendick et al (2011) [Mares, T.E., 2009. An investigation of the relationship between coal and gas properties in the Huntly coalfield, New Zealand, Department of Geological Sciences. University of Canterbury, Christchurch, p. 394.; Papendick, S.L., Downs, K.R., Vo, K.D., Hamilton, S.K., Dawson, G.K.W., Golding, S.D., Gilcrease, P.C., 2011. Biogenic methane potential for Surat Basin, Queensland coal seams. International Journal of Coal Geology 88, 123-134.].

Organic composition (from original plant material) will influence hydrogen content and thus the rate of coal conversion into gas. Also organic composition will influence surface area, which is recognised as a major influence on microbial activity. See: Mares, T.E., Radlinski, A.P., Moore, T.A., Cookson, D., Thiyagarajan, P., Ilavsky, J., Klepp, J., 2009, Assessing the potential for CO₂ adsorption in a subbituminous coal, Huntly Coalfield, New Zealand, using small angle scattering techniques. International Journal of Coal Geology, 77, 54-68; Mares, T.E., Moore, T.A. and Moore, C.R., 2009, Uncertainty in gas saturation in a subbituminous coal seam, International Journal of Coal Geology, 77, 320-327.

Reviewer #2 (Remarks to the Author)

The author cleared up emerged issues. Overall the manuscript is much more easier to understand and of interest to the field of biogenic methanogenesis.

According to my experiences, the in situ scenario will never barely relate to the in vitro scenario and that's why I still think to publish the actually field trial would be of greater interest scientifically. However, from a commercial point of view, I just can't see how the bioreactor-system will be of importance. It would be much more easier and commercial viable to use normal digesters (like the food-waste biogas industry in some countries do) above the surface with a more efficient microbial inoculum resulting in a higher methane turnover as the here used indigenous microbial community.

Reviewer #3 (Remarks to the Author)

The responses to the reviewer comments are complete and reasonable but I would request one addition which is the pump energy for injection. This is a simple calculation and it would ensure

that the process has a positive energy return and is carbon neutral or negative. This would make the paper complete in my view.

Point-by-point response to the reviewers' comments:

Reviewer #1 (Remarks to the Author):

I am fine with the changes made by the authors. it is indeed an interesting paper, though with the caveat from me that I can not evaluate the organic chemistry side of things. My only additional comments on the paper are:

1. Although a stratigraphic section has been given showing the stratigraphy of the sampling area, this hasn't been properly tied into the text. A reference is made in the text to the 'Wyodak' coal zone with the name 'Canyon' given. In fact, the Wyodak coal zone is both the Anderson and Canyon coal seams and their equivalents. Thus in figure 3 the word Wyodak needs to be properly placed for consistency with text. Also, a reference should be given in the text and figure caption for this (the authors might consider the reference (Flores, 2014, Coal and Coalbed Gas: Fueling the Future. Elsevier, Amsterdam, 697 pp. - it is a good place for a lot of the geology which is applicable to their sampling area).

The coal sample was acutely collected from Canyon coal seam (312-321 meters) within Wyodak coal zone. This is made clear in both text and caption of Figure 10. The reference suggested by this reviewer was also added to the caption so that the audiences know where to find more information on the geology aspect of the area.

2. The only response to one of my comments which I am still not fully happy with is their assertion that the organic composition of the coal won't impact on microbial activity. I disagree strongly on that and although not assessed in the experiments discussed in this paper there should at least be some discussion on this topic. The authors should see section 3.1 of Moore (2012) and especially read the thesis of Mares (2009) and the paper by Papendick et al (2011) [Mares, T.E., 2009. An investigation of the relationship between coal and gas properties in the Huntly coalfield, New Zealand, Department of Geological Sciences. University of Canterbury, Christchurch, p. 394.; Papendick, S.L., Downs, K.R., Vo, K.D., Hamilton, S.K., Dawson, G.K.W., Golding, S.D., Gilcrease, P.C., 2011. Biogenic methane potential for Surat Basin, Queensland coal seams. International Journal of Coal Geology 88, 123-134.].

Organic composition (from original plant material) will influence hydrogen content and thus the rate of coal conversion into gas. Also organic composition will influence surface area, which is recognised as a major influence on microbial activity. See: Mares, T.E., Radlinski, A.P., Moore, T.A., Cookson, D. Thiyagarajan, P., Ilavsky, J., Klepp, J., 2009, Assessing the potential for CO₂ adsorption in a subbituminous coal, Huntly Coalfield, New Zealand, using small angle scattering techniques. International Journal of Coal Geology, 77, 54-68; Mares,

T.E., Moore, T.A. and Moore, C.R., 2009, Uncertainty in gas saturation in a subbituminous coal seam, *International Journal of Coal Geology*, 77, 320-327.

The reviewer might misunderstand what we are trying to convey. The coal miners are really concerned on the quality of coal that is proposed to explore for coalbed natural gas. That is why we analyzed the heating values of coal before and after microbiological processes. What we found is that the biological processes do not reduce the heating value of coal (The carbon of this coal was not transformed/degraded). So we conclude that conversion of biomass /carbohydrate within coal by microbes will have no impact on the quality of the coal. I am sure that the characteristics including organic composition, surface area, ect. of coal will greatly impact the microbial activities. However, this is not the major scope of this study.

Reviewer #2 (Remarks to the Author):

The author cleared up emerged issues. Overall the manuscript is much more easier to understand and of interest to the field of biogenic methanogenesis.

According to my experiences, the in situ scenario will never barely relate to the in vitro scenario and that's why I still think to publish the actually field trial would be of greater interest scientifically. However, from a commercial point of view, I just can't see how the bioreactor-system will be of importance. It would be much more easier and commercial viable to use normal digesters (like the food-waste biogas industry in some countries do) above the surface with a more efficient microbial inoculum resulting in an higher methane turnover as the here used indigenous microbial community.

I agree with this reviewer in that anaerobic digesters are easier to produce biogas from biomass; however, the reviewer did not account for the carbon credit. Jim Lucas, Market Development Manager of SoCalGas presented in the 4th Annual Anaerobic Digestion and Biogas Summit 2015, concluded that the carbon credit of low carbon fuel would add value as much as four times to the value of natural gas itself. This is referenced and discussed briefly in the last paragraph of the Discussion. The added value of carbon credit would make this gas economically viable. Moreover, when we talk about coalbed, anaerobic digesters are not comparable in terms of scales. In addition, existing CBNG infrastructure including wells, pipelines, compressors etc. can be re-used. This would save a lot of capital costs.

Reviewer #3 (Remarks to the Author):

The responses to the reviewer comments are complete and reasonable but I would request one addition which is the pump energy for injection. This is a simple calculation and it would ensure that the process has a positive energy return and is carbon neutral or negative. This would make the paper complete in my view.

Pumping energy and CO₂ equivalent were calculated and the numbers were presented in the text (line 382-384).